# Exosomal microRNAs as Biomarkers and Therapeutic Targets for Hepatocellular Carcinoma

**DOI:** 10.3390/ijms22094997

**Published:** 2021-05-08

**Authors:** Andrei Sorop, Diana Constantinescu, Florentina Cojocaru, Anca Dinischiotu, Dana Cucu, Simona Olimpia Dima

**Affiliations:** 1Center of Excellence in Translational Medicine, Fundeni Clinical Institute, 022328 Bucharest, Romania; sorop_andrei@yahoo.com (A.S.); dianaconstantinescu5@gmail.com (D.C.); dima.simona@gmail.com (S.O.D.); 2Department DAFAB, Faculty of Biology, University of Bucharest, 050095 Bucharest, Romania; florentina.cojocaru@unibuc.ro (F.C.); anca.dinischiotu@bio.unibuc.ro (A.D.); 3Digestive Diseases and Liver Transplantation Center, Fundeni Clinical Institute, 022238 Bucharest, Romania

**Keywords:** hepatocellular carcinoma, exosomes, miRNA

## Abstract

Hepatocellular carcinoma (HCC) is the fifth most common cancer worldwide and the second most common cause of cancer-related death globally. This type of liver cancer is frequently detected at a late stage by current biomarkers because of the high clinical and biological heterogeneity of HCC tumours. From a plethora of molecules and cellular compounds, small nanoparticles with an endosomal origin are valuable cancer biomarkers or cargos for novel treatments. Despite their small sizes, in the range of 40–150 nm, these particles are delimited by a lipid bilayer membrane with a specific lipid composition and carry functional information—RNA, proteins, miRNAs, long non-coding RNAs (lncRNAs), or DNA fragments. This review summarizes the role of exosomal microRNA (miRNA) species as biomarkers in HCC therapy. After we briefly introduce the exosome biogenesis and the methods of isolation and characterization, we discuss miRNA’s correlation with the diagnosis and prognosis of HCC, either as single miRNA species, or as specific panels with greater clinical impact. We also review the role of exosomal miRNAs in the tumourigenic process and in the cell communication pathways through the delivery of cargos, including proteins or specific drugs.

## 1. Introduction

Hepatocellular carcinoma (HCC), also known as primary liver cancer, is one of the most severe malignancies worldwide. Recently, the Global Cancer Observatory (GLOBOCAN) reports 840,000 HCC cases and 780,000 individuals per year diagnosed with HCC [1]. Risk factors for HCC include age, hepatitis B or C, cirrhosis, genetic disorders, contaminants, excessive drinking, and smoking. Recently, non-alcoholic fatty liver disease (NAFLD), and its severe form, non-alcoholic steatohepatitis (NASH), were identified as major causes of HCC in western countries, and as being among the main indications for liver transplant after HCV infection [2,3]. NAFLD has been associated with a 2.6-fold increased risk of HCC occurrence [4,5]. Similarly, a study by the US Veterans Health Administration on a group of 1500 patients with HCC showed that HCC patients with NAFLD had a five-fold higher risk of developing HCC in the absence of cirrhosis, compared with HCC and HCV patients [6]. 

Currently, the first-line treatment for small (≤2 cm) HCC tumours is radiofrequency ablation (RFA), which results in reduced disruption and rapid healing, and has comparable effects to surgical resection. An alternative first-line therapy for intermediate-stage patients is transcatheter arterial chemoembolization (TACE), but due to the compensatory effects of vascular proliferation after hypoxia, the treatment efficacy remains unsatisfactory [3]. Sorafenib, the main therapeutic agent for advanced HCC, can substantially extend the average survival time of patients, but it can also enhance tumour progression by inducing drug resistance [7].

The progression of HCC involves the dynamic tumour microenvironment—consisting of fibroblasts, endothelial cells, cancer stem cells, myeloid cells, and related soluble cytokines—impeding early diagnosis of the disease. Therefore, biomarkers and HCC targets are continuously investigated in order to address these unmet needs in clinical practice.

To date, as a non-invasive biomarker in the diagnosis of HCC, clinicians use α-fetoprotein (AFP) most frequently; des-gamma-carboxy prothrombin (DCP) and glypican-3 (GPC3) also serve as non-invasive biomarkers. However, clinical practice revealed that AFP has low sensitivity and specificity for early-stage HCC, as around 50% of patients exhibit negative expression [8]. Better alternatives are imaging procedures, such as magnetic resonance imaging (MRI) and computed tomography (CT), although small tumours cannot be identified using these methods [9]. Histopathological biopsies would complete these diagnosis tools, but they are limited by their invasiveness and the high negative rate of their results [10].

In the last decade, extensive data have described small (nm) extracellular vesicles (EVs) named exosomes as carriers of various molecules—especially microRNAs (miRNA)—and important players in immune and cell–cell communication processes. Hepatic cells generate exosomes in the liver identified in parenchymal cells (hepatocytes), non-parenchymal immune cells (macrophages, dendritic, and T/B natural killer (NK) cells), and non-parenchymal hepatic stromal cells (stellate cells) [11]. The hepatitis B virus (HBV) and hepatitis C virus (HCV) use exosomes to spread viral RNA complexes to neighbouring human liver cells. Furthermore, these particles stimulate nonspecific immune responses through plasmacytoid dendritic cells (PDCs), and suppress unique immune responses by T cells [12]. Due to their creative mechanism of transmitting effector molecules and signals between cells—including RNA, proteins, miRNAs, long non-coding RNAs (lncRNAs), and DNA fragments [13]—exosome EVs have attracted the interest of the scientific community as putative biomarkers, as well as possible delivery vehicles in HCC.

Because the specific isolation of exosomes is challenging, the term exosomes has been indiscriminately used in original and review articles for a heterogeneous population of extracellular vesicles that includes exosomes and other type of vesicles. However, the most consistent results present exosomes as small noncoding RNAs with 19–22 nucleotides and vehicles for miRNAs. We base this review on reports that specify the exosomal nature of miRNAs, and we will use the term hereafter. The miRNAs bind to the 3′-UTR of pre-mRNAs during the post-transcriptional process, and directly target mRNAs for degradation. Today, modern next-generation sequencing and microarray technologies frequently identify miRNAs in the diagnosis of malignant tumours. Several reports of exosomal miRNAs have recently indicated that these particles are better biomarkers for the diagnosis and treatment of HCC than their serum-free-derived counterparts [14]. Exosomal miRNAs exhibit long-term survival due to their better stability in exosomes than in circulating cell-free miRNAs [15]. While exosomes have been explored for several years, we have only recently begun to understand the biological functions of exosomal miRNAs, particularly in HCC.

This review outlines the recent advances in understanding the biological pathways and potential therapeutic applications of exosomes in HCC. We emphasize the role of exosomal content in the tumour microenvironment in HCC and liver metastasis. Additionally, this work addresses the possible use of exosomes as non-invasive biomarkers and as putative therapeutic agents for HCC.

## 2. Biological Characteristics of Exosomes

### 2.1. Exosome Biogenesis and Secretion

Cells release different types of extracellular vesicles (EVs)—including exosomes, microvesicles, apoptotic bodies, ectosomes, and membrane particles—which can be isolated from bodily fluids and from the culture media of the secreting cells [16,17]. According to their biogenesis and size, EVs are divided into three main categories: microvesicles/ectosomes (100 nm–1 µm), released through the plasma membrane; exosomes (30–150 nm), derived from multivesicular bodies (MVBs); and apoptotic bodies (100 nm–5 µm), resulting from cellular apoptosis [18].

Exosomes originating from intracellular endosomes are heterogeneous intraluminal vesicles (ILVs) secreted into the extracellular space. The process of exosome biogenesis has 3 stages: (1) the formation of endocytic vesicles from the plasma membrane; (2) the inward budding of membrane endosomal vesicles, resulting in MVBs; and (3) MVB fusion with the plasma membrane and exosome release (Figure 1).

Briefly, in the first stage, the plasma membrane budding creates the early endosome, and then the endocytic vesicles mature into a late endosome.

In the second stage, membrane inward budding forms intraluminal vesicles (ILVs), which accumulate in the late endosomes and became MVBs. Two pathways mediate MVB formation: the ESCRT (endosomal sorting complexes required for transport)-dependent and ESCRT-independent pathways. The ESCRT consists of four soluble multiprotein complexes—ESCRT-0, ESCRT-I, ESCRT-II, and ESCRT-III—and the VPS4 (vacuolar protein sorting-associated protein 4) complex. Special attention was given to the molecular mechanisms of ESCRT-III/VPS4-dependent membrane remodelling, important in sending membrane budding away from the cytoplasm [19]. Particularly in HCC tissues, one group reported significant downregulation of VPS4A protein expression. This profile is associated with TNM stage, tumour size, tumour capsule integrity, and regional lymph node metastasis [20]. Ectopic overexpression of VPS4A repressed the growth, colony formation, migration, and invasion of HCC cells, whereas the regulation of VPS4A altered the release and uptake of exosomal miRNAs in HCC cells.

ESCRT-independent mechanisms for the formation of ILVs, MVBs, and exosomes include ceramides, cholesterol, D2 phospholipases, and tetraspanins [21,22]. MVBs can be either degraded in lysosomes or released into the extracellular space as exosomes. In the latter case, MVBs move to the cell membrane using cytoskeletal proteins (such as actin and microtubules) [23], motor proteins (dynein, quinine, and myosin) [24], and molecular transducers (some GTPases) [25].

Much effort has been put to unravel the mechanisms of sorting proteins, miRNAs, and other cargos into ILVs, and the process is still not fully understood. A proposed mechanism is sumoylated ribonucleoprotein (HnRNPA2b1), which specifically recognises and internalizes miRNAs in exosomes through binding to specific short motifs [26].

Finally, in the third stage, MVBs fuse with the cell membrane, releasing exosomes into the extracellular space. This process is regulated by the SNARE (soluble NSF-attachment protein receptors) protein complex [27], syntaxin 1A [28], the synaptotagmin protein family, Wnt proteins [29], and Ca^2+^ ions [30]. Rab GTPases—such as Rab7, Rab27a/b, Rab11, and Rab35—mediate the anchoring, docking, and secretion of MVBs to the cell membrane [31,32]. Particularly, Rab GTPases contribute to the secretion of exosomes in HCC cells. One study used HOX transcript antisense RNA (HOTAIR)—a 2158 nucleotide lncRNA transcribed from the HOXC locus [33]. Over-expressed HOTAIR in HCC cells upregulated Rab35 and facilitated the transport of MVBs to the plasma membrane.

Many other proteins and mechanisms participate in exosome biogenesis, and have been well documented in previous review [34,35], but no other HCC-specific processes have been described so far.

### 2.2. Exosome Composition

The nanospheric membrane structure of exosomes resembles that of their parental cells. Some databases (ExoCarta, Vesiclepedia, and EVpedia) contain information about EV composition and procedures for the isolation and purification of these vesicles. The continuous updating of these databases makes them crucial tools for improving the understanding of EVs. ExoCarta’s database gathers the results of exosome composition [36], while Vesiclepedia is more inclusive and provides important information on all categories of extracellular vesicles, constantly updated by the scientific community [37]. EVpedia, a community web portal, includes studies of non-mammalian EVs of different sizes [38]. According to ExoCarta, more than 8000 proteins and 194 lipids are now associated with exosomes [36].

The lipid bilayer resembles the plasma membrane of the cell of origin, containing different types of lipid complexes: sphingomyelin, phosphatidylcholine, phosphatidylethanolamine, phosphatidylserine, monosialotetrahexosylganglioside (GM3), and phosphatidylinositol [39]. Sphingomyelin and GM3 are responsible for the exosome’s rigidity, while phosphatidylserine is expressed on the exosome membrane by different types of phospholipid transport enzymes [40]. Other lipids that support exosome biogenesis are cholesterol, ceramides, and phosphoglycerides.

The exosomal membrane contains the following proteins: integrins, differentiation clusters (CD), tetraspanins, and even the major histocompatibility complex (MHC)-II. Like their parental cells, exosomes contain intracellular proteins, such as the heat shock proteins (Hsp70 and Hsp90), fusion and membrane transport proteins (GTPases, annexins, and flotillins) and tetraspanins (CD9, CD63, CD81, and CD82). These proteins facilitate fusion, cell migration, cell–cell adhesion, and cellular signalling. Adhesion molecules such as integrins mediate cellular binding to the extracellular matrix and to the targeted cells [41]. Exosome-associated proteins include thrombospondin, CD55, CD59, lactadherin, ALIX (ALG-2-interacting protein X), and TSG101 [42], which are incorporated into exosomes during their biogenesis and serve as cargos for cell–cell communication.

Exosomes have tissue-specific molecular content, including DNA, mRNA, miRNA, HSP, and Ras proteins (Rab) [43].

The miRNAs from the cells of origin are to date the most studied constituents as biomarkers. One of the reasons for this is that in different cell types, miRNA species are significantly more abundant in exosomes than in the cell of origin [44]. These findings point toward a specific sorting mechanism of cargo molecules into the exosomes.

Using microarray experiments, previous reports have shown that exosomes derived from mast cells contain a unique set of 1300 mRNAs and 100 miRNAs absent from the cell of origin. In this study, mRNA transcripts from exosomes were translated into the recipient cell [45]. Other reports demonstrated that B cells infected with the Epstein–Barr virus (EBV) secrete mature EBV-miRNA via exosomes. Functional EBV-miRNAs such as BHRF1 and BART are transported to uninfected primary immature monocyte-derived dendritic cells (MoDCs). In these recipient cells, miRNAs trigger functional translational repression of the Epstein–Barr virus latent membrane protein 1 (LMP1) and C-X-C motif chemokine 1 (CXCL11/ITAC) [46].

The calcium-dependent phospholipid-binding proteins (e.g., annexins) play equally important roles in the regulation of RNA loading in exosomes. Annexin-2, the most abundant among annexins, recognizes the 3’-UTR sequence elements of the *MYC* gene, and participates in miRNA packaging in exosomes [47,48,49]. Specific exosomal proteins—such as ALIX, TSG101, flotillin 1, HSP70, CD9, CD81, and CD63—contribute to the protein profile characterization and identification of exosomes obtained through various isolation methods [34].

Exosomes are difficult to isolate and characterize because of their small size, which technically impedes scientists from distinguishing them from other cellular particles. In the following subsection, we review several techniques that are widely used, along with their advantages and disadvantages.

### 2.3. Exosome Isolation and Characterization

The most common exosome extraction method is ultracentrifugation, which allows the separation of large particles and cell debris using a centrifugal force of 200–100,000× *g* [50,51]. However, from the cell culture medium, researchers often extract cell aggregates and other particles [36]. Additionally, this procedure requires a long time, advanced equipment, and high sample volumes. To improve these shortcomings, exosomes are separated in a sucrose density gradient, or with optimal density gradient medium iodixanol solution (OptiPrep), leading to better purification and increased exosome volumes [52,53].

Other common approaches use monoclonal antibodies. Common choices are CD63, CD81, CD82, and CD9 antibodies, epithelial cell adhesion molecules (EpCAMs), and Ras-related protein 5 (Rab5a). These molecules are immobilized under different conditions and combined with magnetic beads and matrix chromatography. To separate them, scientists use plates and microfluidic devices [53]. The drawback of this strategy is that non-exosomal vesicles carrying the antigen also bind to the antibody, thereby reducing the purity of the isolated exosomes. Exosome isolation with ultrafiltration is an appropriate approach, shorter in time than ultracentrifugation, and without the need for specific equipment. Another method of exosome separation is high-efficiency liquid chromatography, which requires advanced technology and results in a sample with high purity [54].

In conclusion, techniques to isolate and purify exosomes are not rigorously adjusted. Therefore, the best approach is to combine isolation with an accurate method for particle characterization. One method is labelling exosome markers—including integrins, tetraspanins (CD81, CD9, and CD63), TSG101, ALIX, and cell adhesion molecules (integrins)—with specific antibodies. Lipid complexes are also sometimes used [43]. Flow cytometry, NTA (nanoparticle tracking analysis), DLS (dynamic light scattering), Western blot, mass spectrometry, and TEM (transmission electron microscopy) are also used in combination with those methods mentioned so far [55].

In conclusion, we cannot identify a gold standard technique for exosome isolation and purification. In the field, several methods employed in combination create a reliable picture of the presence of these particles in bodily fluids. While the isolation and characterization of exosomes are techniques commonly used for all cancers, the function of exosomes, especially miRNAs, is specific to each type of malignancy. The next subsections are dedicated solely to the role of miRNAs in HCC.

## 3. Exosomal miRNA Functions in HCC

### 3.1. Correlation of Exosomal miRNAs with the Clinicopathological Features and Prognosis of HCC

The current incidence and mortality rates of HCC tend to be high. The five-year survival rate for HCC-diagnosed patients is 6%. This unfavourable prognosis stems from the aggressive progression of tumours and high recurrence rates [56,57]. Therefore, early diagnosis is the most important prerequisite for the success of HCC treatment. Exosomes from various sources have different molecular expression profiles, important when comparing the exosome content of patients with those extracted from healthy individuals.

The population of exosomes is relatively simple and stable, transporting different functional molecules—such as proteins, miRNAs, mRNAs, and DNA—to the targeted cells through circulation. Out of all of the above, miRNA species are currently the most investigated molecules as putative biomarkers in HCC therapy, because of their specific signatures and correlation with the presence of disease or with clinical features such as tumour size, disease staging, overall survival, and disease recurrence. Analysis of exosomal miRNA species (mainly from serum) is useful for early detection and assessment of disease progression, without the need for tumour biopsy, which is a major advantage in the diagnosis of liver cancer.

Based on the published data, the role of miRNAs specific to exosomes can be divided into (1) miRNAs specific to HCC patients, (2) miRNAs that significantly correlate with the early stages of the disease, and (3) miRNAs that correlate with tumour staging. Changes in the expression levels of all of these miRNA species derived from exosomes have implications for the diagnosis of hepatic disorders.

Table 1 illustrates different miRNA species extracted from the serum of HCC patients and identified as being clinically significant. miRNA quantification was performed with qRT-PCR. As illustrated, species of miRNA are either over-expressed—as indicated for miR-224, miR-21, miR-210-3p, miR-93, miR-92b, miR-155, and miR-665—or under-expressed—as for miR-718, miR-744, miR-9-3p, and miR-125b—in HCC patients. To this end, we lack explanations about the high variation of miRNA species in HCC patients, which downsizes their specificity for this disease. The crucial message is that exosomal miRNAs are different or better expressed than those extracted from serum, and that some of them have expression correlated with tumour stage or clinical parameters.

The main endpoints in HCC investigation are overall survival (OS) and disease-free survival (DFS); therefore, a direct connection between exosomal miRNAs and patients’ outcomes would be a valuable asset for clinical research.

Table 2 summarizes statistical correlation between the expression of different miRNAs and clinical data. We selected the reports in which the expression levels of miRNAs were obtained following ROC curve analysis, and the diagnostic values were investigated using the area under the curve (AUC) and *p*-values.

Three miRNAs—miR-125b, miR-665, and miR-638—are associated with improved OS rates [65,68,69], whereas exosomal miR-10b-5p overexpression significantly correlates with disease-free survival in patients with HCC [73]. Despite improvements in therapies, the recurrence rates of HCC after surgical resection remain ≥10%. Thus, exploiting exosomal miRNAs (expressed pre- and post-surgery) as predictive markers for recurrence pursues the unmet clinical need.

The expression level of exosomal miR-92b in the serum of HCC patients was significantly higher than that of the control group (non-HCC) [62]. If the levels of exosomal miR-92b continue to be upregulated, premature recurrence is induced. This result identifies exosomal miR-92b as a prognostic biomarker of post-transplant HCC recurrence. Sugimachi et al. showed using microarray analysis that miR-718 is a potential biomarker for predicting HCC recurrence after surgery. The authors reported different expression levels in patients with and without HCC recurrence [66].

Many studies reported low expression of miR-155 in the plasma of HCC patients compared with that of healthy individuals [75,76,77]. Others detected overexpressed miR-155 in preoperative plasma, and found it to be significantly correlated with early recurrence in patients with HCC [64].

A different approach correlates exosomal miRNA expression with tumour staging, which allows for the selection of the most suitable biomarkers for early diagnosis. Overexpression of miR-665 in plasma exosomes correlates with tumour size, invasion, and staging [65], whereas miR-21 expression strongly correlates with tumour stage and cirrhosis. Because exosomal miR-519d and miR-494 are upregulated in HCC compared with liver cirrhosis patients, they are considered to be independent diagnostic biomarkers [70]. In addition, the level of exosomal miR-224 was significantly higher in patients with large tumours (>3 cm) and advanced tumour stages (III/IV) [58]. Similarly, Shi et al. showed that low levels of exosomal miR-638 in HCC were significantly correlated with tumour size (>5 cm) and advanced staging (III/IV) [68].

To gain a better understanding of intercellular communication, and ultimately improve therapeutic strategies for HCC, researchers have recently performed in vitro studies using HCC cell cultures treated with exosomes derived from other cells. HCC cells (Huh7 and SMMC 7721) proliferate significantly less when co-cultured with high concentrations of miR-638 extracted from serum exosomes [68]. Clinical data support these results: low levels of miR-638 extracted from serum exosomes correlate with a lower survival rate of HCC patients. The authors imply that serum exosomal miR-638 affects liver carcinogenesis by inhibiting cancer cell proliferation. In contrast, the expression level of serum-derived exosomal miR-638 did not correlate with viral hepatitis B or C, tumour grade, or liver cirrhosis.

More complex studies were conducted to correlate the type of therapy and development of HCC with the expression of different exosomal miRNAs, based on the putative association of exosomal miRNA species with hepatitis and cirrhosis in HCC patients. The relative expression of exosomal miR-122 (calculated as relative expression of miR-122 after/before TACE) is significantly decreased after TACE [67]. One study was conducted on 57 patients with liver cirrhosis and 18 chronic hepatitis patients with HCC; miR-122 relative expression was higher in patients with liver cirrhosis and longer disease-specific survival. The authors point toward the applicability of miR-122 in the therapeutic guidance of TACE-treated patients.

Other studies suggest that exosomal miR-21 is a potential biomarker for the diagnosis of CHB (chronic hepatitis B) patients. Different reference genes (miRNAs and small RNAs) were used to normalize expression levels of exosomal serum-derived miRs—including miR-221, let-7a, miR-191, miR-26a, and miR-181a—in CHB patients, HCC patients, and healthy individuals. The expression level of exosomal miR-21 was significantly increased in the CHB group compared to the other two groups [78].

Similarly, Murakami et al. analysed miRNA expression profiles from the serum of 64 CHC (chronic hepatitis C) patients and 24 controls with normal liver (NL). The specific expression patterns of exosomal miRNAs expressed in chronic liver disease and inflammation correlated with the types and grades of liver disease. The expression patterns of nine miRNA species (miR-1225-5p, miR-1275, miR-638, miR-762, miR-320c, miR-451, miR-1974, miR-1207-5p, and miR-1246) identified CHC and NL with 96.59% accuracy [79].

Exosomes isolated from HCV J6/JFH-1-infected HuH-7.5 cells, and from the sera of chronic HCV-infected patients, contain the replication enhancers VHC-Ago2, miR-122, and HSP90. Recent studies have consistently shown that miR-122, Ago2, and HSP90 enhance HCV replication. Moreover, miR-122 is a host factor used by HCV for replication, and is present in exosomes isolated from HCV patients [80].

To date, we have identified many miRNA species isolated from exosomes, considering their potential roles as specific biomarkers for HCC. Therefore, scientists have used a more insightful approach of late: studies of miRNA profiling panels that match specific pathological conditions. Panels of miRNAs readily applicable as biomarkers or prognostic tools are yet to be discovered, but studies consider them to be sufficiently accurate to guide therapy in patients with advanced HCC disease. Table 3 reviews in detail these exosomal miRNA panels and their clinical relevance. The exosomes’ sources are indicated.

### 3.2. Effects of Exosomal microRNAs on Cell Survival, Proliferation, and Angiogenesis

HCC cells can release exosomes to promote tumour cell proliferation, angiogenesis, migration, or metastasis (Figure 2).

The two-way signalling pathways impact the drug resistance of HCC. For instance, high levels of miR-744 promote proliferation, and induce resistance of HepG2 cells to sorafenib, targeting paired box gene 2 (*PAX2*), which is overexpressed in HCC tissue. Following treatment with exosomes, miR-744 significantly decreases HCC cell proliferation and resistance to therapeutic drugs [71]. miR-21 induces cell proliferation and metastasis by inhibiting the expression of PTEN [86], PDD4, RECK, and SULF-1 (human sulphatase-1) [87], and confers resistance to chemotherapeutic drugs [88].

In vitro studies indicate that miR-224 decreases the expression of *GNMT* (glycine *N*-methyltransferase) by directly targeting the 3’-UTR mRNA and promoting the proliferation and invasion of HCC cells [58].

Exosomal miR-210 derived from HCC cells is internalized by endothelial cells, and promotes tumour angiogenesis through direct inhibition of the *SMAD4* (SMAD family member 4) and *STAT6* (signal transducer and activator of transcription 6) genes [63]. Matsuura et al. showed that both cellular and exosomal miR-155 expression levels were significantly increased under hypoxic conditions in HCC cells [64]. Of a group of six miRNAs, only miR-155 was upregulated under hypoxic conditions. Another study demonstrated that miR-31 and miR-451 species in exosomes derived from adult human liver stem cells inhibit HCC growth and stimulate apoptosis [89].

The release of miR-26a species in HepG2 cells treated with exosomes led to ectopic overexpression of miR-26a and decreased migration and tumour proliferation. Exosomes loaded with miR-26a blocked cell growth, pointing to the possibility of using engineered exosomes as therapeutic agents [90]. The next subsection of this review notes some results deciphering the mechanism of communication through exosomal miRNAs between HCC cells and cells from the tumour microenvironment.

### 3.3. Role of Exosomal microRNAs in Intra- and Intercellular Communication and Therapies

Exosomes secreted by HCC cells provide autocrine and paracrine signals to surrounding cells but also deliver products to distant cells, fine-tuning diverse biological responses. The process is two-way: blood cells or other cells from the tumoural microenvironment secrete exosomes, which mediate carcinogenesis. All of these processes have been intensively analysed, and many communications pathways have been depicted in recent years.

Of note, even during biogenesis, two miR species (miR-27b-3p and miR-92a-3p) inhibit VPS4A expression in HCC tissues [20]. In turn, dysregulation of VPS4A facilitates the secretion of oncogenic miRNAs in exosomes. The expression of VPS4A in HCC tissue is linked to tumour growth and metastasis.

Exosomal miR-21 derived from HCC cell lines converts normal hepatic stellate cells (HSCs) into cancer-associated fibroblasts (CAFs) through directly targeting the *PTEN* gene, which activates the PDK1/AKT signalling pathway. Finally, angiogenic cytokines (VEGF, MMP-2, MMP-9, FGF2, and TGF) are secreted, which promote cancer progression. These results concur with the clinical data: patients with HCC have high levels of serum exosomal miR-21—correlated with CAF activation, higher vessel density, and lower survival rates [91].

A newly described mechanism shows that miRNAs secreted by HCC cells inhibit T cell function, escaping the immune system. Next-generation sequencing analysis (NGS) reveals that miR-23a-3p has a high expression level in exosomes derived from tunicamycin-treated HCC cells (Exo-TM). Co-cultivation of T cells with macrophages treated with Exo-TM leads to decreased cell ratio of CD8 T and decreased interleukin-2 expression, as a result of the activation of the PTEN/AKT pathway [92].

Drug-resistant HCC cells (Bel/5-FU) secrete high levels of exosomal miR-32-5p, and low levels of the tumour suppressor gene *PTEN*. This discrepancy results from miR-32-5p overexpression, which inhibits PTEN and, subsequently, activates the PI3K/AKT signalling pathway. Inhibited PTEN induces drug resistance, and promotes angiogenesis and EMT. Clinically, miR-32-5p overexpression and low PTEN expression are positively associated with poor prognosis in HCC patients [93].

Exosome-derived miRNAs are mediators of the carcinogenesis induced by environmental chemicals. Exosomes derived from arsenite-transformed L-02 cells transfer miR-155 to normal L-02 and THLE-3 cells (grown in co-culture) and induce pro-inflammatory activity in normal liver cells. Moreover, exosome-derived miR-155 from serum was abundant in the patient group exposed to arsenite [94].

In hepatocytes, miR-155 binds to Toll receptor ligands, triggering Toll receptor-mediated inflammation and promoting liver injury and inflammation [95].

One study identifies new pathways of functional miRNA transfer from circulating tumour cells (CTCs) to the tumoural cells of origin, with the potential to inhibit tumour recurrence and metastatic spread. Exosomal miR-25-5p derived from HCC contributes to “tumour self-seeding”, defined as colonization by CTCs. This mechanism was linked to the expression of the *LRRC7* (Leucine-rich repeat-containing 7) gene involved in CTC reattachment and the reformation of tumour cell seeding. The uptake of exosomes secreted by CTC cells greatly improved the migratory and invasive abilities of HCC cells, with miR-25-5p declared as the main target of *LRRC7*. Inhibition of exosomal miR-25-5p reversed these effects [96].

Exosomal miR-103 produced by hepatic cells and delivered to endothelial cells (HUVECs, LSECs) and normal liver sinusoidal endothelial cells (Sk-Hep-1) inhibited protein junctions, zonula occludens-1, VE cadherin, and p120-catenin. MiR-103 reduced the integrity of endothelial junctions, increased the grade of vascular permeability and, finally, facilitated tumour metastasis [97]. To analyse metastasis and proliferation of target cells, in vitro co-cultivation studies used the transfer of long non-coding RNA (lncRNA FAL1 (Focally amplified lncRNA on chromosome 1)) molecules to HepG2 and HuH-7 cell cultures. The competitive binding to miR-1236 in target cells leads to the overexpression of ZEB1 and AFP, and the subsequent promotion of metastasis and proliferation of the targeted cells [98].

The development of HCC is mainly based on LC disorders, as the tumour microenvironment is enriched with activated fibroblasts. Highly metastatic HCC cells release exosomal miR-1247-3p to convert fibroblasts into CAFs by targeting the *B4GALT3* (β-1,4-galactosyltransferase 3) gene, which activates the β1-integrin/NF-κB signalling pathway. Consequently, the activated pathway decreases the secretion of the IL-6 and IL-8 cytokines. The same study showed that increased expression levels of exosomal miR-1247-3p significantly correlate with lung metastases in patients with HCC [61]. CAFs induce tumourigenic processes, tumour niches, EMT (epithelial–mesenchymal transition), and chemoresistance. All of these processes are important in the development of the inflammatory microenvironment, and promote lung metastases derived from liver cancer. A different group performed sequencing analysis of CAFs and corresponding para-cancer fibroblasts (PAFs), and found a significant reduction in exosomal miR-320a levels in CAF-derived exosomes. The transfer of stromal-cell-derived miR-320a inhibits the tumour progression of HCC cells by binding to its direct target *PBX3*. These results qualify the method of miRNA transfer as a potential treatment option for HCC progression [99].

Human macrophages transfer miRNA species to HCC cells in a manner that requires cell contact through gap junctions. Two specific miRNAs—miR-142 and miR-223—are efficiently transferred; they block the stathmin-1 and IGFR-1 receptors, and finally HCC cell proliferation [100].

Exosomal miR-21, miR-192, and miR-221 derived from colorectal cancer cell lines and transferred to HepG2 and A549 cells promoted the invasion and metastasis of recipient cells [101]. Hep3B-derived exosomes containing specific miRNAs—such as miR-584, miR-517c, miR-378, miR-520f, miR-142-5p, miR-451, miR-518d, miR-215, miR-376a, miR-133b, and miR-367—were internalized in another HCC cell line, HepG2. Consecutively, these miRNAs modulated the TAK1 (TGF-β-activated kinase 1) inhibition process. Loss of TAK1 has been associated with hepatocarcinogenesis [102].

Exosomes released from umbilical mesenchymal stem cells (UMSCs) inhibit HCV infection, especially viral replication. Qian et al. found that the anti-hepatitis C virus infection effect was mediated by various miRNAs (let-7f, miR-145, miR-199a, and miR-221) specifically transported by exosomes derived from UMSCs [103].

The E2 envelope glycoprotein of the hepatitis C virus (HCV-E2) stimulated mast cells to secrete exosomes with high levels of miR-490. The exosome shuttled into HCC cells inhibited the ERK1/2 signalling pathway and, in the end, suppressed HCC cells’ metastasis [104].

Using exosomes to mediate the transfer of miRNAs between different cells encourages clinical application, but for the moment, studies are limited to experiments using cell culture models. Exosomes operate on hepatocytes even when they derive from the cells of other tissues or organs, and contribute to the physiological or pathological processes of the liver [105,106,107,108].

Adipose tissue-derived mesenchymal stem cells (AMSCs) transfected with a plasmid-encoding miR-122 (122-Exo) effectively loaded miR-122 in secreted exosomes. Then, 122-Exo was transported to HCC cells, and altered the expression profiles of cyclin G1 (*CCNG1*), *ADAM10* (a disintegrin and metalloprotease domain-containing protein 10), and *IGFR-1* (insulin-like growth factor receptor 1). These changes occurred after HepG2 cells were exposed to 122-Exo for 24 h [109]. AMSCs infected with pre-miR-199a-3p (LV-199a) plasmid abundantly expressed miR-199, and transferred the species via exosomes after puromycin selection. This process increased the sensitivity of HCC cells to doxorubicin by inhibiting the mTOR signalling pathway. Intravenous injection of an orthotopic HCC murine model with the AMSC-Exo-199a produces a rapid increase in the efficiency of doxorubicin therapy in HCC tumours [110].

Other groups report the use of exosomes as nanotransporters in therapies, paving the way for new applications in HCC treatment. Exosomes transfected with antineoplastic and antifibrotic miR-335-5p were administered in vitro and in vivo to mice with developed HCC tumours; this treatment inhibited cell proliferation, invasion, and tumour growth. Previous reports demonstrated that HSC-derived exosomes can encapsulate and transfer miR-335-5p to HCC cells in vitro or in vivo, inhibiting tumour growth [111]. In a recent study, exosomes were used as vehicles for the transport and targeted internalization of small RNA molecules in HCC cells. HepG2 cells were conditioned to bind (through the Apo-A1 receptor) and internalize exosomes designed by genetic engineering [90].

## 4. Conclusions and Perspectives

Despite many advances in the research of liver diseases to date, the diagnosis of hepatic pathology remains a challenging task. The data reviewed here show that HCC-related exosomes provide great insights for the identification of key molecular factors of HCC. Exosomal secretion from different types of cells, the presence of exosomes in serum of HCC patients, and their involvement in cell–cell communications enable these nanoparticles to play a significant role in both physiological and pathological processes. In clinical application, exosomes and other EVs provide an enriched source of miRNAs as compared to serum-free miRNAs, and contain a large amount of tRNA fragments, which are promising novel biomarkers for diagnosis.

Regardless of the significant potential of exosomes, there are still many important problems. To this end, accurate isolation and characterization of exosomes is still the subject of scientific debate, which opens new directions for studies focused on the development of reagent kits. Techniques for isolating small extracellular vesicles should be standardized and improved before being implemented in the clinic, as current methods are time consuming, and only small amounts of biological material are obtained. Future analyses should elucidate the pathways that can affect HCC development and progression or effector functions for exosomal miRNA species.

Additionally, the comprehensive mechanisms of exosomes in HCC invasion and metastasis are still unclear, which impedes their application in HCC diagnosis and treatment. Several studies of the diagnosis and treatment of HCC are still in the preclinical stage, and clinical applications require more data-based biomarker screening.

A recent review presents promising results of engineered exosome-based therapeutics. Exosomes containing interferon genes (STING) or Il-12 are in phase I/II clinical trials on patients with lymphoma. Different approaches and different exosome sources are being tested [112].

The huge amount of miRNA species continuously isolated and characterised suggests at a first glance that we are far from pointing to a specific exosomal biomarker in HCC.

MiRNAs are extremely heterogeneous in different EVs and exosomes; therefore, a comprehensive inventory of these cargos and EV subclasses would be of great help [57]. We also know that a percentage of miRNAs and mRNAs are lost to degradation or are functional in cell communication. Therefore, a clear delineation of these classes and processes is the bridge to clinical translation.

Based on preclinical results and clinical trials, we envisage the development of engineered exosomes with optimized carriers and miRNAs able to break the tumorigenic process. With the efforts of the scientific community toward establishing reference genes, the application of validation methods and the use of bioinformatical analysis enforce the hope that we are close to opening the new era of exosomal biomarkers.

## Figures and Tables

**Figure 1 ijms-22-04997-f001:**
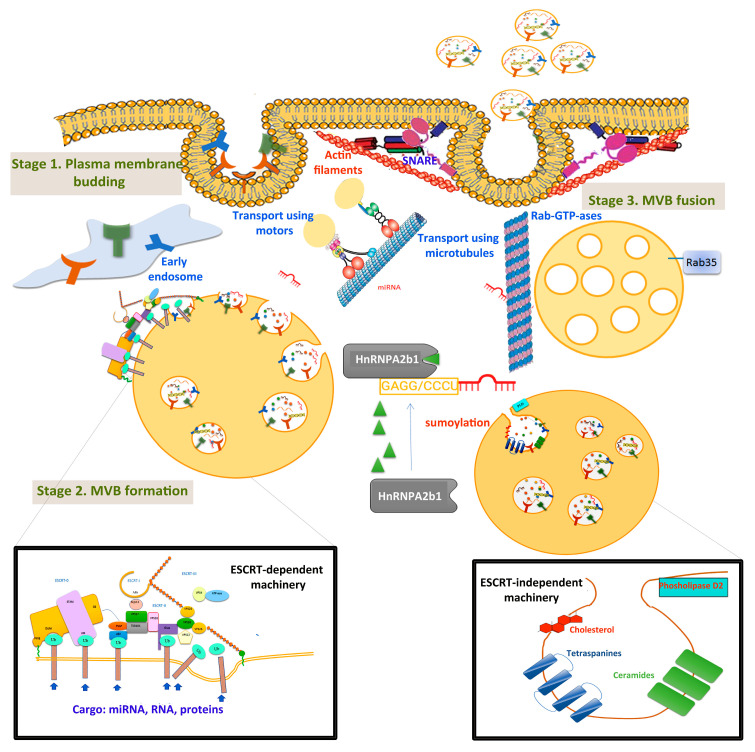
Extracellular vesicles’ biogenesis, transport, and membrane fusion. The first stage, membrane budding, integrates receptors that are transferred into early endosomes. MVB formation (stage 2) and sorting take place in the late exosome. Detailed ESCRT-dependent and ESCRT-independent pathways are presented in insets. MVBs are transported via microtubules and motor proteins, and fuse with the cellular membrane using SNARE, synaptotagmin proteins, and actin filaments. The pathway proposed for miRNA sorting using sumoylated HnNPA2b1 protein is also illustrated.

**Figure 2 ijms-22-04997-f002:**
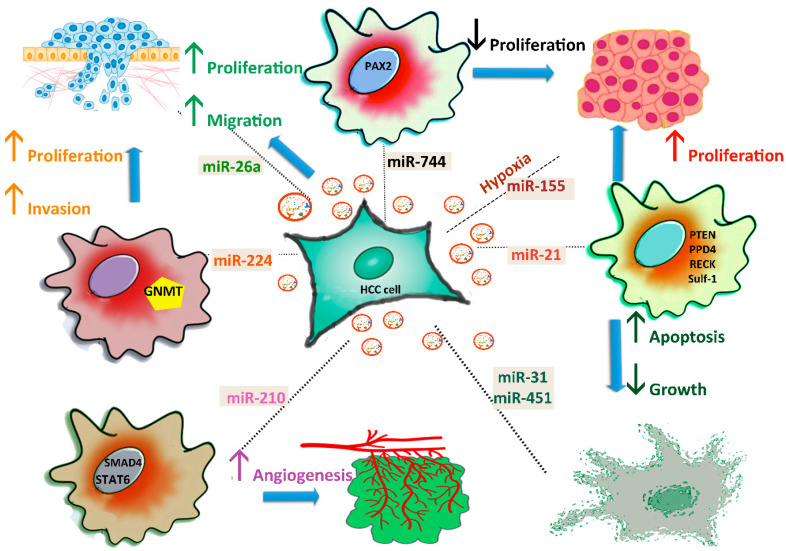
Roles of exosomal miRNAs in HCC progression. HCC cells secrete exosomes and affect proliferation by transferring miR-744, miR-21, and miR-26a into neighbouring cells. Angiogenesis is stimulated by miR-210, which blocks the SMAD4 and STAT6 pathways, while apoptosis is activated by miR-31 and miR-451. Inhibition of GNMT by miR-224 promotes the proliferation and invasion of HCC cells. miR-155 secretion increases under hypoxic conditions.

**Table 1 ijms-22-04997-t001:** Exosomal miRNAs with clinical significance in HCC.

miR Species	Expression Profile in HCC Down- or Upregulated	Exosome Isolation Methods	Groups and Sample Size	Normalized (Reference miRs/Internal Control)	Clinical Significance	References
miR-224	Up	Total Exosome Isolation Kit	HCC (*n* = 89) and healthy controls (*n* = 50)	let-7i, let-7g, and let-7d	Detection, prognosis, recurrence, and therapuetic target	[58]
miR-21		Total Exosome Isolation Reagent (serum)	HCC (*n* = 30), CHB (*n* = 30), and healthy controls (*n* = 30)	U6	Detection and diagnosis	[59]
miR-93		Total Exosome Isolation kit	HCC (*n* = 85) and healthy controls (*n* = 23)	miR-16	Detection, prognosis, and recurrence	[60]
miR-1247-5p		Ultracentrifugation	HCC patients without lung metastasis (*n* = 90), HCC patients suffering lung metastasis (*n* = 20), and healthy controls (*n* = 25)	18S	Detection, diagnosis, and therapuetic target	[61]
miR-92b		ExoQuick Exosome Precipitation Solution	Non-HCC (*n* = 26), HCC no recurrence (*n* = 28), early HCC recurrence (*n* = 43), and late HCC recurrence (*n* = 22)	Synthetic C. elegans miR-39	Detection, prognosis, and recurrence	[62]
miR-210-3p		Ultracentrifugation	HCC (*n* = 104) and healthy controls (*n* = 60)	cel-miR-67 (NC67)	Detection, diagnosis, and therapuetic target	[63]
miR-155		ExoQuick Exosome Precipitation Solution (System Biosciences)	HCC (*n* = 40): high level miR-155 of HCC (*n* = 20) and low level miR-155 of HCC (*n* = 20)	Not mentioned	Prognosis and recurrence	[64]
miR-665		ExoQuick TM Kit (System Biosciences)	HCC (*n* = 30) and healthy controls (*n* = 10)	U6	Detection and prognosis	[65]
miR-718	Down	Ultracentrifugation	Patients with HCC who underwent living donor liver transplantation (LDLT) (*n* = 59)	Synthetic C. elegans miR-39	Detection, prognosis, recurrence, and therapeutic target	[66]
miR-122		ExoQuick	Samples collected before and after TACE treatment of HCC patients (*n* = 75).	cel-miR-39	Detection and prognosis	[67]
miR-638		Total Exosome Isolation kit	HCC (*n* = 126) and healthy controls (*n* = 21)	miR-16	Detection, prognosis, and recurrence	[68]
mi-125b		ExoQuick Exosome Precipitation Solution	Group 1: CHB (*n* = 30), LC (liver cirrhosis) (*n* = 30), and HCC (*n* = 30). Group 2: HCC (*n* = 128)	cel-miR-39	Detection, prognosis, and recurrence	[69]
miR-9-3p		Ultracentrifugation	HCC (*n* = 30) and healthy controls (*n* = 10)	not mentioned	Potential treatment/therapeutic target	[70]
miR-744		Ultracentrifugation	Group 1 (serum):HCC patients (*n* = 10) and healthy controls (*n* = 10). Group 2 (tissue): HCC patients (*n* = 68) and normal liver tissue (*n* = 52)	U6	Detection, diagnosis, and therapeutic target	[71]

**Table 2 ijms-22-04997-t002:** Exosomal miRNAs with statistically significant expression with respect to clinical data.

No.	miR Species	Diagnostic Values (AUC Values, Area under the Curve)		Correlation with Prognosis and/or with Clinicopathological Features (*p* Values)			References
		AUC Values (>0.7)	*p* Values	Large Tumour Size (>3 cm or 5 cm)	Advanced Tumour Stage (III/IV)	Kaplan–Meier Curve Analysis: OS and DFS	
1	miR-224	0.910 (95% CI: 0.84–0.98)	*p* < 0.001	*p* < 0.001	*p* < 0.001	*p* < 0.01	[58]
2	miR-93	0.825 (95% CI: 0.730–0.919)	*p* < 0.0001	*p* = 0.047	*p* = 0.006	*p* = 0.046	[60]
3	miR-92b	0.702 (95% CI: 0.576–0.828)	*p* = 0.004	ND	ND	ND	[62]
4	miR-665	ND		*p* = 0.0042 (<5 cm)	*p* = 0.0276	*p* < 0.05	[65]
5	miR-718	ND		*p* = 0.04	*p* = 0.026	*p* = 0.0002 (DFS)	[66]
6	miR-125b	0.739 (95% CI: 0.648–0.830)	*p* = 0.048	*p* = 0.11	*p* = 0.011	*p* < 0.001 (DFS and OS)	[69]
7	miR-194	0.738 (95% CI: 0.638–0.838)	*p* = 0.0001	*p* = 0.013	*p* > 0.05	ND	[72]
	miR-17-5p	0.850 (95% CI: 0.764–0.936)	*p* = 0.0001	*p* = 0.047	*p* > 0.05	ND	
	miR-106a	0.704 (95% CI: 0.534–0.873)	*p* = 0.016	*p* = 0.035	*p* > 0.05	*p* = 0.041	
8	miR-10b-5p	0.968 (95% CI: 0.85–0.99)	*p* < 0.0001	ND	not statistically significant		[73]
	miR-215-5p	0.936 (95% CI: 0.80–0.99)	*p* < 0.0001	ND	*p* < 0.01	*p* = 0.02 (DFS)	
9	miR-595	0.92 (95% CI: 0.86–0.97)	*p* < 0.0001	ND	*p* = 0.007	ND	[74]

Abbreviations: overall survival (OS); disease-free survival (DFS); receiver operating characteristic (ROC); confidence interval (CI); not described (ND).

**Table 3 ijms-22-04997-t003:** Exosome panels and their clinical relevance.

No.	miR Analysed in Panels	Expression Profile in HCC (Up- or Downregulated)	Exosome Isolation Methods	Source of Exosomes	Sample Size and Groups	Quantification Methods	Normalized (Reference miRs/Internal Control)	Clinical Relevance	References
1	miR-122, miR-125b, miR-145, miR-192, miR-194, miR-29a, miR-17-5p, and miR-106a.	up	Total Exosome Isolation Kit (GenePharma)	Human serum	HCC (*n* = 80) and healthy controls (*n* = 30)	qRT-PCR	miR-16	Distinguish HCC patients from healthy controls.	[72]
2	miR-18a, miR-221, miR-222, miR-224	up	ExoQuick Exosome Precipitation Solution		HCC (*n* = 20), CHB (*n* = 20), and LC (*n* = 20)		miR-16	Distinguish HCC from LC and CHB.	[81]
	miR-101, miR-106b, miR-122, miR-195	down							
3	miR-26a, miR-29c, miR-21	down	ExoQuick reagent	Human serum	HCC (*n* = 20), CHB (*n* = 20), and LC (*n* = 20)		let-7a	Lower levels in HCC patients than in cirrhotic and HBV patients.	[82]
4	miR-122, miR-148a and miR-1246	up	Polyethylene glycol (PEG) 6000 (Sigma-Aldrich, St Louis, MO) of 8% concentration		Discovery group: HCC (*n* = 5) with LC (*n* = 5) for deep sequencing. Validation group: NC (*n* = 64), CHB(*n* = 50), LC (*n* = 53), and HCC(*n* = 68) for further study 85 HCC patients		cel-miR-39	Distinguish early-stage HCC from liver cirrhosis.	[83]
5	miR-21, miR-10b	up	Ultracentrifugation	Rat serum	Male fisher 344 rats (*n* = 108) (HCC models)		miRNA-484	Biomarkers for early-stage HCC combined with exosomal miRNAs and AFP.	[84]
	miR-122, miR-200a	down							
6	miR-10b-5p, miR-18a-5p, miR-215-5p, and miR-940	up	SeraMir Exosome RNA Amplification Kit (System Biosciences)	Human serum	HCC (*n* = 90), CHB (*n* = 27), LC (*n* = 33), and healthy controls (*n* = 28)	qRT-PCR and miR sequencing for discovery set	miR-1228-3p	Biomarker for early-stage HCC and poor disease-free survival	[73]
7	miR-519d, miR-21, miR-221 and miR-1228	up	Ultracentrifugation		30 patients (10 patients with liver cirrhosis without nodular liver lesions, 13 patients with early HCC, and 7 patients with advanced HCC) and HCC-derived cell lines (*n* = 7). HepG2, Hep3B, Huh-7, SNU449, SNU398, SNU182, and SNU475	qRT-PCR	cel-miR-39	More efficient diagnostic role than AFP for HCC patients.	[74]
8	miR-140-3p, miR-30d-5p, miR-29b-3p, miR-130b-3p and miR-330-5p	up	3D Medicine exosome isolation kit (CFDA license no. Hu min xie bei 20170019)		10 fast-migrated and 10 slow-migrated PDC cultures from 36 HCC samples	Small RNA library construction and sequencing	Normalization	Migratory abilities of tumour cells	[85]
	miR-296-3p	down							

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
