# Peer review of "Exosomal microRNAs as Biomarkers and Therapeutic Targets for Hepatocellular Carcinoma"

_ijms, 2021, doi:10.3390/ijms22094997_

Round 1

Reviewer 1 Report

  1. It clearly stated exosomal miRNAs are possible biomarkers for clinical diagnosis and prognosis of HCC.
  2. Minor spellings: please use one word like either miRNA or microRNA throughout paper. 
  3. Line 255 and 256 :miRNA155 and MIR155.
  4. For table 2, suggest add one more column as species of sample source, like human or animal, because it will more clearly indicate which miRNA biomarker would be confirmed in animal model.
  5. Line 168-170, please provide source.

Author Response

We thank the reviewer for the comments and after carefully reading the  we made the following changes:

  1. We added some paragraphs in the conclusion and perspective" section to highlight the importance of exosomes as biomarkers and possible therapeutics.
  2. We track and replace microRNA with miRNA
  3. We made the correction
  4. We added one more column with source indication. Table 2 became table 3.
  5. We added the reference.

Reviewer 2 Report

The manuscript by Sorop et al. constitutes a review manuscript on the role of exosomal microRNAs as biomarkers and therapeutic targets for hepatocellular carcinoma. The authors herein provided a very good state-of-the-art and critical review on this topic, highlighting the most important information in the field and raising the most important concerns. However, there are some points that should be addressed to increase the overall quality of this work.

Major concerns:

1- The term exosomes has been extensively and indiscriminately used in original and review articles although it is known that the specific isolation of exosomes is challenging. The most used techniques isolated a population of extracellular vesicles that include exosomes and other type of vesicles. Therefore, I strongly encourage the authors to discuss this point and to consider to alter the term exosomes to “extracellular vesicles”.

2- What would be the benefit of measuring exosome miRs as a diagnostic tool, when compared with serum-free miRNAs? Some discussion on this topic should be added, focusing on the future translation of the findings.

3 - Since this paper is mostly focused on exosomes/EVs, a figure with the biogenesis of EVs would be of value.

4 – Regarding the miR exosomes as potential diagnostic and prognostic biomarkers, apart from the tables already included, it would be of great value to include the main diagnostic values (AUC, sensitivity, specificity) of the mentioned studies in a new table. Similarly (or in the same table), the information of correlation with prognosis and correlation with clinicopathological findings would be great.

5 – Regarding the effect of exosomal miRs in cell survival, proliferation and angiogenesis, a summary figure with the main effects of the miRs in HCC pathogenesis would be of value.

6 – Subsection 3.3 should be re-organized since it does not follow a logical order. I strongly suggest to firstly mention the effects of exosomes secreted by HCC cells and then the other way around (exosomes secreted by other cell types).

7 – Some discussion on the most promising biomarkers and therapeutic targets should be included in conclusions. Please discuss what should be the next steps in order to translate some of the findings into clinics.

 Minor concerns:

1- In introduction, the authors mention the main causes of HCC without mentioning metabolic liver diseases (NAFLD) which are currently regarded as the major cause of HCC worldwide. Please include it.

2- Et al, in vitro, in vivo, gene names and species should be in italics in all the manuscript and in the tables.

3 – In line 348, the authors mention miR-155 and in the next sentence, they refer to miR-15a. Is this a typo?

Author Response

We thank the reviewer for the constructive suggestions that surely improved our manuscript. Ou

Major concerns:

  • It is true that is difficult to discriminate between these two terms, and mostly it is difficult to take for granted that all the vesicles from different studies are exactly “nanovesicles”. However, we selected those papers referring to” exosomal microARN” and all the review is constructed around this concept. We decided not to completely change the title and the main text, but however to address the problem correctly noted by the reviewer (rows 69-75).
  • The main advantages of exosomal miRNAs would be protection offered for transport and enrichment of exosomes with specific miRNas. Apparently some other species (tRNA) are also exosome specific and this idea was added at row 466-467. Translation to clinic is complicated by the heterogeneity of cargos (different microARNs) and different vesicle types. Moreover, some miRs are degraded in recipient cells. A clear “catalogue” to classify these EVs in subclasses would be a first step (as suggest by a previous review). Also, some preclinical results presented recently in Nature are encouraging. The problem addressed in Conclusion and Perspectives rows 480-490.

3 -We added Figure 1 which illustrates biogenesis and traffic conform to cited papers.

4 –We added a new table (table 2)  former table 2 became Table 3,  and we thank for this suggestion. It brings a new value to our MS.

5 Figure 2 illustrates the summary of  section 3.2.

6 – We re-organized the section 3.3, hopefully in a more user-friendly manner.

7 – We make some suggestions based on the preclinical reports of several biotech companies reported in Nature (ref. nr. 112)

 Minor concerns:

  • We did so, thank you for the suggestion it brings some novelty to the HCC epidemiology.
  • We tracked the expression and made the changes.

3 – No, it is not a typo, it is a different story not very related to the previous one. I deleted.

Round 2

Reviewer 2 Report

All my concerns were addressed and this review is ready do be published. Congratulations on this work.